# Clinical Presentation of Community-Acquired Legionella Pneumonia Identified by Universal Testing in an Endemic Area

**DOI:** 10.3390/ijerph17020533

**Published:** 2020-01-15

**Authors:** Shruti Puri, Monique Boudreaux-Kelly, Jon D. Walker, Cornelius J. Clancy, Brooke K. Decker

**Affiliations:** 1Division of Infectious Diseases, Medical University of South Carolina, Charleston, SC 29425, USA; puri@musc.edu; 2Statcore, VA Pittsburgh Healthcare System, Pittsburgh, PA 15240, USA; Monique.Kelly@va.gov (M.B.-K.); Jon.Walker4@va.gov (J.D.W.); cjc76@pitt.edu (C.J.C.); 3Division of Infectious Diseases, University of Pittsburgh, Pittsburgh, PA 15261, USA

**Keywords:** *Legionella*, Legionnaires’ disease, *Legionella* pneumonia

## Abstract

The rapid identification of *Legionella* pneumonia is essential to optimize patient treatment and outcomes, and to identify potential public health risks. Previous studies have identified clinical factors which are more common in *Legionella* than non-*Legionella* pneumonia, and scores have been developed to assist in diagnosing cases. Since a *Legionella* pneumonia outbreak at VA Pittsburgh in 2012, nearly all patients with pneumonia have been tested for *Legionella*. The purpose of this study was to evaluate distinguishing characteristics between *Legionella* and non-*Legionella* pneumonia with the application of universal testing for *Legionella* in all cases of community-acquired pneumonia. We performed a retrospective case-control study matching *Legionella* and non-*Legionella* pneumonia cases occurring in the same month. Between January 2013 and February 2016, 17 *Legionella* and 54 non-*Legionella* cases were identified and reviewed. No tested characteristics were significantly associated with *Legionella* cases after Bonferroni correction. Outcomes of *Legionella* and non-*Legionella* pneumonia were comparable. Therefore, in veterans who underwent routine *Legionella* testing in an endemic area, factors typically associated with *Legionella* pneumonia were non-discriminatory.

## 1. Introduction

Legionnaires’ disease is a pneumonia caused by *Legionella* bacteria associated with case fatality rates of 10–25% [1]. *Legionella* can account for up to 15% of cases of community-acquired pneumonia and has been associated worldwide specifically with human-made water systems [1,2]. These water systems can include, but are not limited to, cooling towers, hot tubs, decorative fountains, shower heads, and medical equipment [1]. Various studies have identified clinical and laboratory factors that are associated with *Legionella* pneumonia [2,3,4,5,6,7,8,9,10]. These prediction models were derived from pneumonia cases diagnosed without mandatory testing for *Legionella*. Indeed, current guidelines do not recommend routine testing for *Legionella* [11]. Rather, targeted testing of healthcare and severe community-acquired pneumonias is recommended—an approach that misses a significant percentage of *Legionella* cases [12]. Therefore, studies of *Legionella* pneumonia have artifactual biases toward more severe cases and presentations with conventionally attributed symptoms.

Following a Legionnaires’ disease outbreak at the Veterans Affairs Pittsburgh Healthcare System (VAPHS) [13], we instituted mandatory *Legionella* testing for all patients with pneumonia [14]. *Legionella* was subsequently identified in 1% of patients with pneumonia; almost half of these cases would have been unrecognized if we followed current testing guidelines. Our systematic protocol afforded an opportunity to identify characteristics of *Legionella* pneumonia in an endemic area, without biases introduced by selective testing. The purpose of this study was to identify the differentiating characteristics between *Legionella* and non-*Legionella* pneumonia in patients with pneumonia.

## 2. Materials and Methods

After Institutional Review Board approval was obtained (IRB Pro00001725), a retrospective, case-control comparison of patients with community-acquired *Legionella* pneumonia or non-*Legionella* pneumonia at VAPHS was performed. *Legionella* pneumonia was identified from January 2013 through February 2016—a period in which >97% of pneumonia cases were tested for *Legionella* [14]. No outbreak cases have been identified at VAPHS since 2012. Inclusion criteria for community-acquired pneumonia were clinical features of pneumonia along with radiographic findings on chest radiography or another imaging modality. These criteria were based on Centers for Disease Control, (Atlanta, Georgia) surveillance definitions [11,15]. One control patient was excluded as *Legionella* testing was not performed. Patients with pneumonia were tested by *Legionella* urinary antigen (BinaxNOW^®^
*Legionella* Urinary Antigen Card, Alere, Waltham, MA, USA) and/or sputum culture for *Legionella*, as previously described [14]. Patients were diagnosed with *Legionella* pneumonia if any of urinary antigen, sputum culture, or autopsy lung culture were positive for *Legionella*. For each month when there was at least one *Legionella* pneumonia case diagnosed, all the non-*Legionella* pneumonia cases from that month were randomized into a list and the first five were then chosen. Demographic and clinical factors, disease severity, and outcomes were compared. Chi-square and Fisher exact tests were used as appropriate to determine statistical significance. After Bonferroni correction [16] for 44 variables, a calculated value of *p* < 0.001 was considered significant.

## 3. Results

Overall, 1691 pneumonia cases were diagnosed. Seventeen cases of *Legionella* pneumonia (1%) occurred in 11 of 38 months; none of these cases were hospital-acquired. Fifty-five patients with non-*Legionella* pneumonia were randomly identified and 54 were included in the analysis (representing 3.2% and 11% of pneumonias overall and in reviewed months, respectively).

Twelve and nine patients with *Legionella* pneumonia had positive urinary antigens and sputum cultures, respectively. One patient was negative by urinary antigen and sputum culture, but an autopsy lung biopsy culture grew *Legionella pneumophila*. Fifteen and two patients were infected with *L. pneumophila* serogroups 1 and (2–14), respectively. Fifty-one (93%) and 37 (67%) non-*Legionella* pneumonia patients had negative *Legionella* urinary antigens and sputum cultures, respectively. One patient in the non-*Legionella* pneumonia group had neither urinary antigen nor sputum sent for *Legionella* testing.

Demographic information and clinical data are shown in Table 1 and Table 2, respectively. 

## 4. Discussion

To our knowledge, this is the first study assessing factors associated with *Legionella* pneumonia in a clinical setting that employed universal testing. Immunomodulatory therapy, fever, tachycardia, hyponatremia and proteinuria trended toward association with *Legionella* pneumonia by univariate analysis. Nursing home residency and sputum production trended toward an association with non-*Legionella* pneumonia. However, none of these factors were present in a majority of patients with either type of pneumonia, nor were the factors significant after Bonferroni correction. In the end, clinical and laboratory factors previously found to be predictive of *Legionella* pneumonia either were not significantly different in the two groups or they were not tested consistently (e.g., C-reactive protein, lactase dehydrogenase, Creatine phosphokinase (CPK) [3,4,5,6,7,8,9,10,11].

None of the prior studies of *Legionella* pneumonia risk factors or prediction models utilized a Bonferroni correction, which likely would have reduced or eliminated the significance of findings [1,2,3,4,5,6,7,8,9,10]. We did not retrospectively apply the Winthrop University weighted score [9] or the Ito-Ishida score [17], as none of our patients had majority of the components included in each. In order to utilize these tools accurately, clinicians must perform tests that are not typically performed for community-acquired pneumonia. Rather than conducting these additional tests and applying a score, we found it simpler in veterans from an endemic area to send *Legionella* testing.

In our population, the percentage of patients with a class IV or V pneumonia severity index was higher among patients with non-*Legionella* pneumonia than those with *Legionella* pneumonia (72% and 59%, respectively), and patients’ outcomes were comparable. The data demonstrate that *Legionella* pneumonia presents across a spectrum of severity. Studies identifying factors that predicted *Legionella* pneumonia or reporting increased mortality and morbidity compared to other community-acquired pneumonias [2,3,4,5,6,7,8,9,10] may have been biased by a failure to test all patients for *Legionella* and the inclusion of only more severe cases or patients with previously identified risk factors.

Current guidelines do not recommend routine *Legionella* testing since empirical treatment regimens for pneumonia generally include antibiotics with anti-*Legionella* activity [11], and testing in low-risk settings may not be cost-effective [18]. In regions that are endemic for *Legionella*, however, systematic testing identifies cases that would otherwise remain undetected [12,14,19], facilitates targeted antibiotic therapy, and serves a public health function as surveillance for potential outbreaks. Rapid, accessible microbiologic methods are more sensitive and specific than any scoring algorithm and may be easier to use as a screening method than a complex score requiring multiple tests not typically sent on non-critically ill pneumonia patients. In the aftermath of an outbreak, such as occurred in our hospital [13], routine *Legionella* testing afforded confidence that cases were not missed, and that water management and infection prevention protocols remained effective. 

There are limitations to this study. Our study population was ≥95% male. Results in our patients may not be relevant to other cohorts, since veterans are typically older and have more health conditions than the general population [20]. Veterans may also be hospitalized sooner as they have a higher use of medical resources compared to the general population [20]. *Legionella* is known to be more prevalent in the northeastern part of the country, and in Pittsburgh in particular, than in other regions [21]. In addition to differences in clinical findings or characteristics, patients with *Legionella* pneumonia in other parts of the country may be detected at later, more severe stages of the disease. The relatively small number of patients we encountered and our single center study design may bias our findings due to low statistical power. Finally, since this was a retrospective study, we were limited to laboratory or clinical data that were collected clinically. Although some factors were considered significant by univariate analysis before the Bonferroni calculation, we felt it was important to use the correction to avoid the promotion of weak associations of limited practical utility or potentially erroneous associations unique to our patient population.

## 5. Conclusions

*Legionella* pneumonia in our experience was indistinguishable from other causes of community-associated pneumonia. Our experience suggests that testing all pneumonia cases for *Legionella* in endemic areas is more practical than attempting to derive prediction models to identify high-risk patients for testing. The broad application of testing in an endemic area allows for better-targeted antibiotic therapy and may increase public health awareness and help prevent outbreaks with heightened infection control practices as a result of the testing. In non-endemic regions, testing should be considered on a case-by-case basis while bearing in mind the indistinguishable characteristics between *Legionella* and other causes of community-associated pneumonia.

## Figures and Tables

**Table 1 ijerph-17-00533-t001:** Demographic characteristics of pneumonia cases. N: Number, w/in: Within, HIV: Human Immunodeficiency Virus, COPD: Chronic Obstructive Pulmonary Disease. *p*-Values obtained by Fisher exact or Chi-square testing as appropriate.

	Non-*Legionella*	*Legionella*	*p*-Value
	N = 54 (%)	N = 17 (%)	
Male	51 (94)	17 (100)	1
Age > 75	17 (31)	1 (6)	0.11
Age 50–75	35 (65)	15 (88)	
Age < 50	2 (4)	1 (6)	
Steroid therapy	2 (4)	2 (13)	0.24
Nursing home resident within last 30 days	39 (72)	2 (13)	0.031
Hospitalization w/in last 90 days	23 (43)	4 (24)	0.16
Enrolled in Wound Care	2 (4)	0	1
HIV	0	0	
Cancer or hematologic malignancy	17 (31)	4 (24)	0.76
Chemotherapy in last 6 months	3 (5)	1 (6)	1
Immunomodulator/biologic therapy	2 (4)	3 (18)	0.085
Current tobacco use	16 (29)	10 (59)	0.082
Any tobacco use	37 (69)	14 (82)	0.57
Active alcohol abuse	3 (6)	0	1
Diabetes mellitus	17 (31)	5 (29)	1
COPD	33 (61)	6 (35)	0.093
Heart failure	15 (28)	5 (29)	1
Kidney disease	13 (24)	4 (24)	1
Cirrhosis	1 (2)	2 (13)	0.14
Cerebrovascular disease	10 (19)	1 (6)	0.28

**Table 2 ijerph-17-00533-t002:** Characteristics of *Legionella* and pneumonia cases. AMS: Altered Mental Status, PSI: Pneumonia Severity Index, RR: Respiratory Rate, SBP: Systolic Blood Pressure, AST: Aspartate Aminotransferase, ALT: Alanine Aminotransferase, O2: oxygen. *P*-values obtained by Fisher exact testing.

Clinical Characteristics	Non-*Legionella*	*Legionella*	*p*-Value
	N = 54 (%)	N = 17 (%)	
Cough	34 (63)	12 (71)	0.77
Shortness of breath	30 (56)	7 (41)	0.41
AMS/Confusion	18 (33)	6 (35)	1
Sputum production	27 (50)	5 (29)	0.17
Complaint of diarrhea	7 (13)	2 (12)	1
Myalgia/arthralgia	11 (20)	5 (29)	0.75
PSI Class 1–3	16 (30)	7 (41)	0.38
PSI Class 4–5	39 (72)	10 (59)	
Infiltrate	54 (100)	17 (100)	
Uni-lobar infiltrate	33 (61)	14 (82)	0.15
Multi-lobar infiltrate	21 (39)	3 (18)	0.15
Pleural effusion	17 (31)	5 (29)	1
Vital signs on admission			
RR > 29	3 (6)	3 (18)	0.14
SBP < 90	7 (13)	1 (6)	0.67
Temp > 39.0 Celsius	5 (9)	7 (41)	0.0055
Pulse > 124	2 (4)	3 (18)	0.085
Laboratory characteristics			
pH < 7.35	4 (7)	2 (12)	1
Serum phosphorus < 2.5 mg/dL	5 (9)	5 (29)	0.25
Serum sodium < 133 mEq/L	7 (13)	8 (47)	0.0055
Urine protein > 30	7 (13)	8 (47)	0.0094
AST > 42 IU/L	13 (24)	5 (29)	1
ALT > 60 IU/L	8 (15)	2 (12)	0.71
Glucose > 249 mg/dL	5 (9)	3 (18)	0.39
Hematocrit < 30%	7 (13)	2 (12)	1
Lactate dehydrogenase > 180 IU/L	0	1 (6)	0.11
Creatine kinase > 200 U/L	4 (7)	4 (24)	0.63
C-reactive protein > 1.0 mg/dL	2 (4)	1 (6)	1
Partial pressure O2 < 60	6 (11)	5 (29)	0.42
Platelet count < 171 × 10^9^/L	15 (28)	7 (41)	0.28

No variables were significant after Bonferroni correction. Mortality was low in our cohort and not significantly different between groups at 30 days, 90 days, or after hospital discharge (Table 3). All patients received *Legionella*-active treatment on presentation with pneumonia.

**Table 3 ijerph-17-00533-t003:** Patient outcomes by group. ICU: Intensive Care Unit. Successful treatment refers to resolution of pneumonia clinical symptoms. *p*-Values obtained by Fisher exact testing.

	Non-*Legionella*	*Legionella*	*p*-Value
	N = 55 (%)	N = 17 (%)	
ICU admission	18 (33)	10 (59)	0.088
Successful treatment	51 (94)	15 (88)	0.59
Survival to discharge	51 (94)	15 (88)	0.59
30-day survival	43 (80)	15 (88)	0.72
90-day survival	39 (72)	15 (88)	0.21

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
