# Peer review of "Clinical Presentation of Community-Acquired Legionella Pneumonia Identified by Universal Testing in an Endemic Area"

_ijerph, 2020, doi:10.3390/ijerph17020533_

Round 1
Reviewer 1 Report
This paper looks at Legionella pneumonia in veterans at a Pittsburg health center. However the results are very narrow to a single center and discussion needs to be expanded for wider application.
Practically and critically speaking, how does this work link to the national health guidance or other best practices for Legionella pneumonia detection? Are there any global studies in a different context, and how do the results from Pittsburg apply to the public health issues globally. Otherwise, it is far too narrow in scope and literature for audience interest. Please add the IRB protocol number into Section 2. Line 115. Please set the context better for this paper in the discussion. Why is this region endemic for Legionella? Is it environmental, social or other reasons? Please add as much detail as possible, as this will also link better to other similar contexts. Line 123. Study limitations, consider adding that veterans may have better access to health care because of their government benefits of service which means patients with pneumonia may seek care earlier than a non-insured patients. Table 1 and 2. There is a lot of acronyms that are very specific to this medical niche, please though write these out in full for reader benefit. COPD, PSI Class, RR, SBP, others. Expand the conclusions. How could this work be implemented for better public health practices? At least give some practical options for solutions in this demographic. The authors have given ‘polite’ recommendations only that need to be critically and constructively expanded to be useful. How much would expanding to 100% testing cost? What can be done with resource limited environments? Author contributions. There is a lot of spelling and formatting problems, please check this section.
Author Response
Thank you for your comments.
Practically and critically speaking, how does this work link to the national health guidance or other best practices for Legionella pneumonia detection?
This work suggests that guidance that does not endorse universal testing may bias the syndrome of the disease to those cases that are tested for. Legionella disease is missed. Universal testing (as we perform) may be the best way to identify Legionella disease in an endemic setting.
Are there any global studies in a different context, and how do the results from Pittsburgh apply to the public health issues globally.
Other studies have shown that Legionella is missed. (see references 12,14, 19)
Otherwise, it is far too narrow in scope and literature for audience interest. Please add the IRB protocol number into Section 2.
Thank you, this has been added.
Line 115. Please set the context better for this paper in the discussion. Why is this region endemic for Legionella? Is it environmental, social or other reasons? Please add as much detail as possible, as this will also link better to other similar contexts.
It is not known what makes an area endemic for Legionella, however, many authors have speculated and found associations with humidity, temperature and precipitation. References are included in the cited line, beyond that, this speculation is outside the scope of this paper.
Line 123. Study limitations, consider adding that veterans may have better access to health care because of their government benefits of service which means patients with pneumonia may seek care earlier than a non-insured patients.
Thank you, this has been added.
Table 1 and 2. There is a lot of acronyms that are very specific to this medical niche, please though write these out in full for reader benefit. COPD, PSI Class, RR, SBP, others.
Thank you, this has been added.
Expand the conclusions. How could this work be implemented for better public health practices? At least give some practical options for solutions in this demographic. The authors have given ‘polite’ recommendations only that need to be critically and constructively expanded to be useful. How much would expanding to 100% testing cost? What can be done with resource limited environments? Author contributions. There is a lot of spelling and formatting problems, please check this section.
Thank you, the conclusions were expanded. In our hospital, we diagnose about 540 cases of Pneumonia every year. Our Legionella urinary antigen test costs approximately $24 per test, thus we spend $12,960 per year on Legionella testing, or about $2993 per patient identified (average 4.3 cases over the last 6 years)
Reviewer 2 Report
Thank you for your contribution.
It seems important to provide a rationale for conducting this comparison. Legionella's disease appears to be related to problematic water and cooling systems within an institution such as a hospital. Therefore, readers need to understand the additional benefit from comparing persons with Legionella's disease and persons with pneumonia over a certain time period.
The rationale for choosing a case control study may need to be detailed. Similarly, it is important to indicate the inclusion, exclusion criteria for the sample and the factors that were identified from the medical record to conduct the case control study.
In the results, the first table uses a distinction of percentage/N. All other tables are just the other way round: n/percentage. It seems important to have the same presentation in all tables.
Please indicate in the abstract, introduction and discussion the rationale for conducting this study in the first place.
Author Response
Thank you for your review
It seems important to provide a rationale for conducting this comparison. Legionella's disease appears to be related to problematic water and cooling systems within an institution such as a hospital. Therefore, readers need to understand the additional benefit from comparing persons with Legionella's disease and persons with pneumonia over a certain time period.
Thank you. We have added a line in the abstract and introduction highlighting this rationale.
The rationale for choosing a case control study may need to be detailed. Similarly, it is important to indicate the inclusion, exclusion criteria for the sample and the factors that were identified from the medical record to conduct the case control study.
A case-control was chosen as the best way to compare patients in our population. As we perform universal testing as a policy after an outbreak in our facility, we would not be able to randomize patients to not being tested. We have updated the methods to include the exclusion and inclusion criteria. All factors identified from the medical record are shown in tables 1 and 2.
In the results, the first table uses a distinction of percentage/N. All other tables are just the other way round: n/percentage. It seems important to have the same presentation in all tables.
Thank you, we apologize for the oversight. This has been standardized.
Please indicate in the abstract, introduction and discussion the rationale for conducting this study in the first place.
Thank you, we have added statements to this effect in those areas.
Reviewer 3 Report
Comments
This study performed a retrospective case-control study matching Legionella and non-Legionella pneumonia cases. Several factors were significantly associated with Legionella pneumonia and non-Legionella pneumoniaafter univariate analysis, but none of these factors was present in a majority of patients with either type of pneumonia. Thus, the authors suggests that testing all pneumonia cases for Legionella in endemic areas is more practical than attempting to derive prediction models to identify high-risk patients for testing.
Some suggestions are listed as follows:
(1) Table 1 and Table 2 and Table 3, Percentage (N) or N (Percentage)? they should be consistent.
(2) Factors with p < 0.05 should be marked with star (*)
Author Response
Table 1 and Table 2 and Table 3, Percentage (N) or N (Percentage)? they should be consistent.
We apologize for the oversight, this has been corrected.
(2) Factors with p < 0.05 should be marked with star (*)
Thank you, but do to the use of Bonferroni correction (reference 16) for multiple variables, only a p<0.001 would be considered significant. No variables were significant in our analysis.
Reviewer 4 Report
The paper aims to assess the applicability of current testing guidelines for Legionella pneumonia. The authors conducted a retrospective case-control study in an endemic area to avoid selective testing biases. They showed that routine Legionella pathogen testing is more reliable than clinical factors to prevent missed Legionella cases.
The authors clearly state the weaknesses and limitations of the study, but without mentioning that almost all included patients were men and therefore making the results not entirely applicable to the normal population.
Specific comments referring to line numbers, tables or figures.
It would make the tables easier to follow if the total number of included patients would be indicated at the top of the table.
Table 1: for “current tobacco use” the p-value is missing.
Table 2: for “Multi-lobar infiltrate” the p-value is missing.
Table 2: have patients with diabetes been included in the parameter “Glucose > 249 mg/dL”?
In table 1, the values show first the percentage and then in brackets the total number, in table 2 it is vice versa. It would be more convenient for the reader if this would be the same in both tables.
The title of table 3 explains the abbreviation N for numbers, whereas this is not explained in the titles of tables 1 and 2, although this abbreviation is used. Since this is a known and common abbreviation, it can be left out.
In all tables, it should be indicated, which statistical test was used to calculate the obtained p-value.
Line 63: How were the non-Legionella patients randomly identified?
Lines 70-71: was the one patient with neither urinary antigen nor sputum sent for Legionella testing still included in the study? If yes, how could this patient be assigned to a study population without knowing if he is Legionella-positive?
Discussion:
Line 90: amongst others, fever and tachycardia were significantly associated with Legionella pneumonia. Could it be possible, that tachycardia in these patients was caused by the fever and therefore is no direct parameter for Legionella pneumonia?
Line109: Which studies are meant? References needed.
Lines 113-114: when empiric treatment regimens already include antibiotics active for Legionella, why is specific testing needed? Would it not be more cost-effective to just treat all patients in endemic areas with antibiotics active for Legionella?
Author Response
Thank you for your review.
It would make the tables easier to follow if the total number of included patients would be indicated at the top of the table.
Thank you, we have made this improvement.
Table 1: for “current tobacco use” the p-value is missing.
Table 2: for “Multi-lobar infiltrate” the p-value is missing.
Thank you, these have been added. Due to the Bonferroni correction, neither of these values are significant.
Table 2: have patients with diabetes been included in the parameter “Glucose > 249 mg/dL”?
All patients who had a glucose obtained that was over 249. This does include patients with diabetes. On review on the data, only 2 of the 8 patients with glucose over 249 were not previously diagnosed as diabetic.
In table 1, the values show first the percentage and then in brackets the total number, in table 2 it is vice versa. It would be more convenient for the reader if this would be the same in both tables.
Thank you, this has been fixed.
The title of table 3 explains the abbreviation N for numbers, whereas this is not explained in the titles of tables 1 and 2, although this abbreviation is used. Since this is a known and common abbreviation, it can be left out.
Thank you, this has been corrected.
In all tables, it should be indicated, which statistical test was used to calculate the obtained p-value.
Thank you, this has been added.
Line 63: How were the non-Legionella patients randomly identified?
The method by which we randomized has been clarified in the methods. Basically, we randomized the list of patients and selected the first 5.
Lines 70-71: was the one patient with neither urinary antigen nor sputum sent for Legionella testing still included in the study? If yes, how could this patient be assigned to a study population without knowing if he is Legionella-positive?
Thank you, we included this patient as he was randomly selected as one of the pneumonia cases from that month. Testing for Legionella was ordered on this patient but was not collected. We have excluded the patient from the analysis and recalculated the statistics.
Discussion:
Line 90: amongst others, fever and tachycardia were significantly associated with Legionella pneumonia. Could it be possible, that tachycardia in these patients was caused by the fever and therefore is no direct parameter for Legionella pneumonia?
Per our use of the Bonferroni correction, these findings were not significantly associated, but were trending toward association. This has been clarified in the manuscript. We agree that fever and tachycardia are unlikely to be independent of each other, but have been reported independently in other studies. In any case, the point we intended in this manuscript was that symptoms and findings are not reliably associated with Legionella disease in an endemic population.
Line109: Which studies are meant? References needed.
The references have been added, thank you.
Lines 113-114: when empiric treatment regimens already include antibiotics active for Legionella, why is specific testing needed? Would it not be more cost-effective to just treat all patients in endemic areas with antibiotics active for Legionella?
Thank you for your comment. Even though empiric regimens include Legionella treatment, identification of the organism may allow for narrowing treatment and for public health investigation of potential reservoirs of this preventable pathogen. This is addressed in Lines 133-135 and the conclusions section.
Round 2
Reviewer 1 Report
Thank you for the invitation to review this revised manuscript. I commend the authors’ effort in revising and improving the quality of the paper. As much as I would like to support this research to be indexed, I still have significant concern about the quality of the paper.
One comment of mine in the first round was skipped:
Table 1 and 2. There is a lot of acronyms that are very specific to this medical niche, please though write these out in full for reader benefit. COPD, PSI Class, RR, SBP, others.
Author Response
Table 1 and 2. There is a lot of acronyms that are very specific to this medical niche, please write these out in full for reader benefit. COPD, PSI Class, RR, SBP, others.
Thank you. These were added to the table explanation above the respective tables.
Reviewer 4 Report
Thank you for the detailed and well-executed revision of the manuscript.
Author Response
Thank you for the detailed and well-executed revision of the manuscript.
Thank you, we very much appreciate your thoughtful review that improved our manuscript.